# Diagnostic accuracy of a sequence-specific *Mtb*-DNA hybridization assay in urine: a case-control study including subclinical TB cases

Yves Tschan,[1,2] Mohamed Sasamalo,[3] Hellen Hiza,[1,2,3] Jacques Fellay,[4,5,6] Sébastien Gagneux,[1,2] Klaus Reither,[1,2] Jerry Hella,[3] Damien Portevin[1,2]

**ABSTRACT** Tuberculosis (TB) caused by *Mycobacterium tuberculosis* (*Mtb*) remains one of the deadliest infectious diseases globally. Timely diagnosis is a key step in the management of TB patients and in the prevention of further transmission events. Current diagnostic tools are limited in these regards. There is an urgent need for new accurate non-sputum-based diagnostic tools for the detection of symptomatic as well as subclinical TB. In this study, we recruited 52 symptomatic TB patients (sputum Xpert MTB/RIF positive) and 58 household contacts to assess the accuracy of a sequence-specific hybridization assay that detects the presence of *Mtb* cell-free DNA in urine. Using sputum Xpert MTB/RIF as a reference test, the magnetic bead-capture assay could discriminate active TB from healthy household contacts with an overall sensitivity of 72.1% [confidence interval (CI) 0.59–0.86] and specificity of 95.5% (CI 0.90–1.02) with a positive predictive value of 93.9% and negative predictive value of 78.2%. The detection of *Mtb*-specific DNA in urine suggested four asymptomatic TB infection cases that were confirmed in all instances either by concomitant Xpert MTB/RIF sputum testing or by follow-up investigation raising the specificity of the index test to 100%. We conclude that sequence-specific hybridization assays on urine specimens hold promise as non-invasive tests for the detection of subclinical TB.

**IMPORTANCE** There is an urgent need for a non-sputum-based diagnostic tool allowing sensitive and specific detection of all forms of tuberculosis (TB) infections. In that context, we performed a case-control study to assess the accuracy of a molecular detection method enabling the identification of cell-free DNA from *Mycobacterium tuberculosis* that is shed in the urine of tuberculosis patients. We present accuracy data that would fulfill the target product profile for a non-sputum test. In addition, recent epidemiological data suggested that up to 50% of individuals secreting live bacilli do not present with symptoms at the time of screening. We report, here, that the investigated index test could also detect instances of asymptomatic TB infections among household contacts.

**KEYWORDS** tuberculosis, diagnosis, non-sputum, urine, cell-free DNA

With the exception of the severe acute respiratory syndrome coronavirus 2 pandemic, tuberculosis (TB) has remained the deadliest human disease caused by a single infectious agent worldwide for decades (1). In 2015, the World Health Organization (WHO) "End TB strategy" aimed to halve TB incidence by 2025 (1, 2). With an 8.8% reduction at the end of 2022, it is evident that more substantial investments into the development of new diagnostics, treatments, and vaccines are needed to better control TB (3). TB is transmitted via aerosols containing live *Mycobacterium*

Address correspondence to Damien Portevin, damien.portevin@swisstph.ch.

The authors declare no conflict of interest.

See the funding table on p. 10.

*tuberculosis* (*Mtb*). In the past, only symptomatic cases were thought to contribute to the spread of TB. However, active case finding suggests that up to 50% of individuals excreting *Mtb* are asymptomatic (4, 5). As a consequence, such individuals could also contribute to TB transmission through mere tidal breathing and without the need for coughing (6, 7). Actually, the treatment of cases identified through repeated community-wide screening had a greater impact than passive case findings on the prevalence of pulmonary tuberculosis at the end of the intervention (8). Thus, there is an urgent need to include the diagnosis and management of asymptomatic TB cases in routine TB control strategies.

Among current diagnostic tools for active TB, sputum-smear microscopy lacks sensitivity and requires further molecular or culture-based testing for confirmation. *Mtb* culture remains the gold standard for TB detection, but time to culture positivity requires several weeks, making it suboptimal for rapid detection and control of TB (9, 10). In 2011, the Xpert MTB/RIF (Xpert; Cepheid, Sunnyvale, CA, USA), a nucleic acid amplification test, was endorsed by the WHO as a new diagnostic tool for pulmonary and extrapulmonary TB (EPTB) (11). In 2015 and 2016, the urine-based lateral-flow lipoarabinomannan (LAM) and the sputum-based loop-mediated isothermal amplification TB assays were approved by WHO, respectively, as alternatives to sputum-smear microscopy for the detection of pulmonary TB (12, 13). The sensitivity of LAM-based diagnostics in urine is inversely proportional to the patients' CD4 cell counts and is solely recommended for advanced HIV-infected individuals with low CD4 counts (14–16). Compared to sputum microscopy, Xpert MTB/RIF has excellent specificity and increased sensitivity and thus has been progressively replacing sputum microscopy as a first-line rapid diagnostic tool (11). However, particular patient populations, namely infants and immunocompromised individuals, are often unable to produce sputum and/or harbor paucibacillary infections. In addition, sputum analysis in itself becomes irrelevant in instances of EPTB. Consequently, in 2014, the WHO established a target product profile (TPP) for the development of a non-invasive biomarker and non-sputum-based diagnostic tool that is applicable at the community level, affordable, and implementable in resource-limited settings (17). In the meantime, a series of non-sputum-based diagnostic tools have been developed and recently reviewed (18). With a sensitivity of 93.4% and a specificity of 100%, the detection of LAM in exhaled breath condensates has given promising results (19). In the area of blood-based biomarkers, we previously demonstrated that starting from a single milliliter of blood, T cell activation markers can be used to diagnose TB in adults and children with a sensitivity and a specificity close to the optimal TPP targets (20, 21). Moreover, a recent meta-analysis showed that blood transcriptional signatures overall display moderate sensitivity for the detection of incipient TB (22). Xpert MTB/RIF has been reported to detect *Mtb*-DNA in clinical specimens other than sputum as well as in low-volume sputum and other salivary samples (23, 24). Cell-free *Mtb*-DNA may be detected in urine samples (25) independently of HIV infection status (26–28). Compared to blood or breath condensates, urine collection is non-invasive and technically non-challenging, and therefore, constitutes a particularly attractive non-sputum-based specimen for TB diagnosis. However, urine cell-free DNA fragments are particularly short, labile, and diluted, which lowers the sensitivity of methods based on standard DNA extraction or PCR protocol design (25–31). A hybridization approach enabling concentration of *Mtb*-specific (IS6110) transrenal DNA fragments substantially increased the sensitivity of *Mtb* cell-free DNA testing of urine specimens (32, 33). In that context, and following the diagnostic accuracy reporting standards (STARD) (34), we assessed the accuracy of a sequence-specific hybridization assay that detects the presence of *Mtb* cell-free DNA in urine in a case-control study that included asymptomatic TB cases.

## MATERIALS AND METHODS

### Study population

After informed consent, TB patients with a positive sputum Xpert MTB/RIF were recruited into the TBDAR cohort study conducted in the Temeke District of Dar es Salaam, Tanzania. The institutional review board of the Ifakara Health Institute, the Medical Research Coordinating Committee of the National Institute of Medical Research, and the Ethikkommission Nordwest- und Zentralschweiz in Switzerland approved the TBDAR study protocol related to the present study. The TBDAR cohort study enrolls TB patients as well as household contacts (HHCs). Eligibility criteria for HHCs were no previous records of a positive TB test or history of TB treatment. Patients and HHCs underwent similar clinical examinations and questionnaires and on the same day, sputum and urine specimens were collected for further laboratory analysis. In the case of patients, specimens were collected prior to the initiation of antibiotic treatment. Anonymized clinical data and laboratory tests' results as well as laboratory report forms were centrally stored within the Open Data Kit Central secured server hosted by the Swiss TPH (35). Clinical data encompassing participants' characteristics, symptoms, and Xpert MTB/RIF results from sputum were used to stratify participants by disease status and presence of symptoms after performing and interpreting the index test.

### Reference standard

Xpert MTB/RIF on sputum specimen was used as a reference standard to determine sensitivity and specificity of the index test. The Xpert MTB/RIF test was performed at the National Tuberculosis and Leprosy Program laboratory. For household contacts, the collected sputum specimens were transferred to the TB laboratory of the Ifakara Health Institute, Bagamoyo branch and subjected to Xpert MTB/RIF Ultra testing following Cepheid's instructions.

### Index test

The index test was adapted from Oreskovic et al. (32). Urine specimens collected in parallel with sputum samples were treated within 10 min of collection by transferring 10 mL into tubes pre-filled with 500 µL 0.5 M EDTA, pH 8.0 (Sigma-Aldrich, St. Louis, USA) and 100 µL 1 M Tris-HCl, pH 8 (Thermo Fischer, Waltham, USA) before cooled transport (4°C) to the laboratory and stored at −80°C. On the day of analysis, urine specimens were randomly sorted in batches and relabeled so that scientists performing the index test were blinded from clinical characteristics and laboratory test results. Positive and negative template controls (NTCs) were realized using commercially acquired pooled urine specimens from healthy individuals (Lee Biosolutions, Maryland Heights, USA) that were spiked or not in-house with synthetic positive control (SPC) template. Dynabeads MyOne Streptavidin C1 (Thermo Fischer, Waltham, USA) (50 µL per urine sample) were washed thrice with an equal volume of high salt wash buffer (1 M NaCl, 10 mM Tris-HCl pH 8, 0.05% Tween-20) before rotating incubation with 50 µL of magnetic beads containing 25 pmol of each capture probe for 15 min at room temperature. Capture beads were washed thrice and resuspended in an equal volume of high salt buffer. Urine specimens (~10 mL) from patients and household contacts were processed in monoplicates while positive controls were carried out in duplicates or triplicates. Urine samples were thawed and centrifuged for 10 min at $4{,}347 \times g$ and supernatants transferred into 15 mL tubes before adding 50 µL of capture beads. Specimens were incubated at 120°C for 15 min in a dry bath and place under rotation at room temperature for 30 min before centrifugation for 10 min at $4{,}347 \times g$. Supernatants were discarded and beads transferred within left-over fluid into 1.5 mL DNase-free tubes and dried after standing 1 min on a magnetic rack before subsequent washes twice with 1 mL of high salt wash buffer and once with 1 mL of low salt wash buffer (15 mM NaCl, 10 mM Tris-HCl pH 8). Captured DNA targets were eluted after addition of 20 µL of molecular grade NaOH 20 mM and subsequent neutralization with 3.5 µL of HCl 0.1 M.

## Primers, probes, synthetic positive control, and qPCR protocol

Primers, capture probes, and SPCs were purchased from IDT, Coralville, USA with sequences as described previously (36). qPCR reactions were conducted on a CFX96 Touch Real-Time PCR Detection System (Bio-Rad Laboratories, Hercules, CA, USA) and its associated Bio-Rad CFX Maestro Software v5.3. The reaction mix consisted of 5 µL DNA template or urine extraction output, 1× HOT FIREPol EvaGreen qPCR Supermix (Solis Biodyne OÜ, Tartu, Estonia), and 0.2 µM of forward and reverse primers in a final volume of 50 µL topped up with certified nuclease-free water. The qPCR protocol consisted of an initial 12-min denaturation step at 95.0°C, followed by 45 amplification cycles of 95°C for 30 s, 64°C for 30 s, and 68°C for 1 min. The PCR phase was followed by a melt curve analysis from 65°C to 95°C in increments of 0.2°C every 5 s with an initial denaturation phase of 95°C for 30 s. Cycle threshold ($c_t$) values were determined by the qPCR software at a threshold of 500 RFU.

## Statistical methods

Study participants were recruited consecutively aiming for a sample size of $n = 100$ with equal representation of patients and household contacts. Statistical analysis encompassing paired $t$-test, Wilcoxon test, and linear regression was carried out with GraphPad Prism v8.1.2 (GraphPad Software, San Diego, USA). When specified, statistical significance was corrected for multiple comparisons using the Holm-Sidak method. Receiver operating characteristics (ROC) analysis, including confidence interval calculations and plots, was performed using the R package pROC.

## RESULTS

### Index test yield and cut-off determination

We applied an *Mtb*-specific DNA capture approach (32) on urine specimens from healthy individuals spiked with $10^1$–$10^4$ copies of a synthetic positive control and compared the resulting qPCR calibration curve to qPCR reaction mixes containing a comparable amount of spiked DNA. Linear regression for both calibration curves revealed comparable excellent goodness of fit ($R^2 > 0.96$) (Fig. 1A). The mean $c_t$ difference between real-time amplification of directly spiked DNA and those from DNA captured and eluted through the hybridization procedure was 0.46. This indicates that in average, 72.6% of spiked DNA could be recovered from the capture bead assay. Furthermore, negative template controls performed on a pool of urine specimens from healthy individuals were repeatedly and independently subjected to the index test ($n = 7$, Fig. 1B). The lower limit of the 95% confidence interval (CI) (38.91–43.88) of the cycle threshold values obtained for the NTCs was used to establish the cut-off for positivity/negativity determination of the index test results at a $c_t$ of 38.91. This cut-off, after extrapolation from the linear regression model equation, was equivalent to 4.2 copies of target DNA per 10 mL of urine (Fig. 1B).

### Index test accuracy results

The clinical characteristics of the TB patients and household contacts recruited between 31 January and 23 November 2022 are summarized in Table 1. All patients had at least two symptoms including cough while only six HHCs reported cough, of which one also suffered from weight loss, chest pain, and fever. Complying with current standards for reporting diagnostic accuracy studies (34), a STARD diagram of the study profile is presented in Fig. 2 to list the processed specimen stratified by index test and reference standard results. For quality control purposes, positive and negative template controls were processed in parallel with patients' specimens for each run of index test assessment. Out of 112 study participants, urine samples from 110 could be blindly subjected to the index test (STARD diagram, Fig. 2). The processed specimens included 52 samples from TB patients and 58 samples from HHCs. Results from 22 samples were excluded

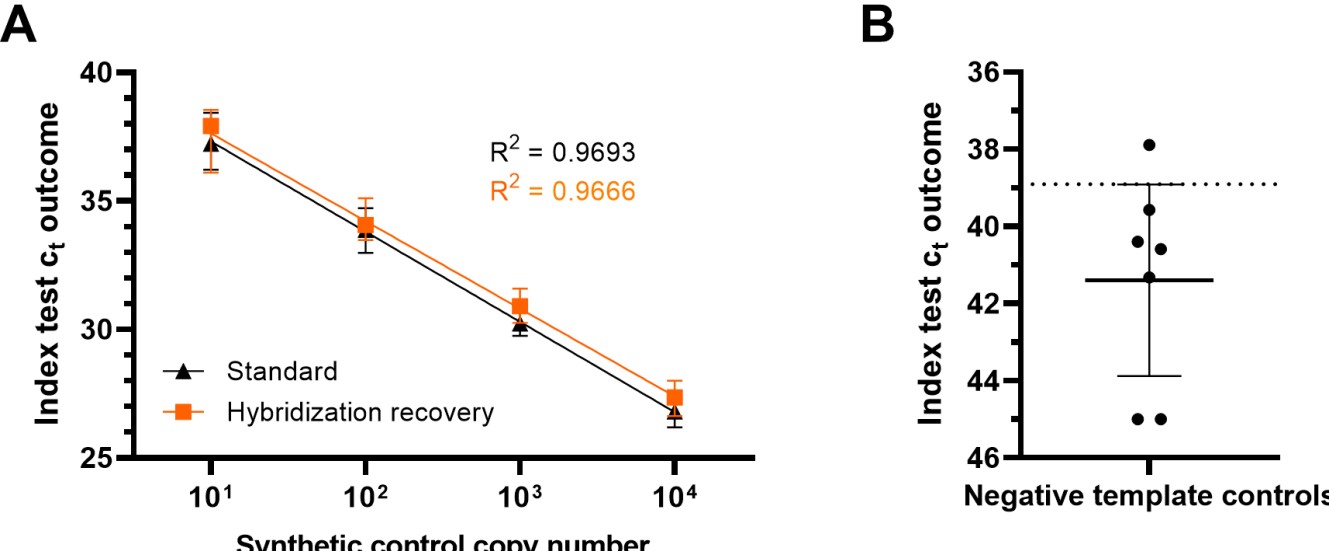

**FIG 1** Index test yield and boundaries assessment. (A) Calibration curves resulting from index test results over four dilution series of water templates (black) spiked with $10^1$–$10^4$ copies of SPC or eluted target DNA from equally spiked urine samples that had been subjected to the hybridization capture protocol (orange). The resulting $R^2$ values from a linear regression model are indicated. Target DNA copies = $10^{(c_t - 41.04)/-3.411}$. (B) Urine specimens from healthy individuals were subjected to the index test to determine the lower limit of the 95% CI (38.91–43.88) as the index test cut-off equivalent to 4.2 copies of the target DNA sequence.

(13 Xpert MTB/RIF positive TB patients and 9 HHC samples) due to technical errors (master mix preparation, $n = 10$) or experimental caveats (magnetic beads aggregation, $n = 9$ or cell debris carryover, $n = 3$). Overall, the index test results were completed from 39 symptomatic Xpert MTB/RIF positive TB patients and 49 HHCs, of which four displayed positive Xpert MTB/RIF test results (Fig. 4A). Index test quantitative results for Xpert MTB/RIF positive participants ranged from 0 to 2,999 copies per 10 mL of urine (Interquartile range 1.96–375.0; median 28.26) and were significantly enriched in the *Mtb*-specific target DNA compared to Xpert MTB/RIF negative participants (Fig. 3A, $P > 0.0001$). At a threshold of 4.83 determined by the Youden's index, ROC analysis of the index test resulted in an area under the curve of 0.86, corresponding to a sensitivity of

**TABLE 1** Demographics and clinical characteristics of patients and household contacts at the time of study enrolment

|  | TB patients ($n = 53$) | Household contacts ($n = 59$) |
| --- | --- | --- |
| Age, median (IQR) | 36.2 (26.7–47.3) | 36.8 (27–45) |
| Female, $n$ (%) | 13 (24.5) | 34 (57.6) |
| Xpert MTB/RIF positive[a], $n$ (%) | 53 (100) | 5 (8.5) |
| Symptoms, $n$ (%) |  |  |
| Cough | 53 (100) | 6 (10.2) |
| Weight loss | 43 (81.1) | 1 (1.7) |
| Chest pain | 39 (73.6) | 1 (1.7) |
| Fever | 31 (58.5) | 1 (1.7) |
| Night sweat | 8 (15.1) | 0 (0) |
| Culture, $n$ (%) |  |  |
| Positive for MTBC[b] | 23 (43.4) | 2 (3.4) |
| Negative | 28 (52.8) | 47 (79.7) |
| Positive for NTM[c] | ND | 7 (11.9) |
| Contaminated | 2 (3.8) | 3 (5.1) |

[a]No errors or invalid results among negatives, rifampicin resistance detected in one household contact.
[b]MTBC, *Mycobacterium tuberculosis* complex.
[c]NTM, non-tuberculous mycobacteria.

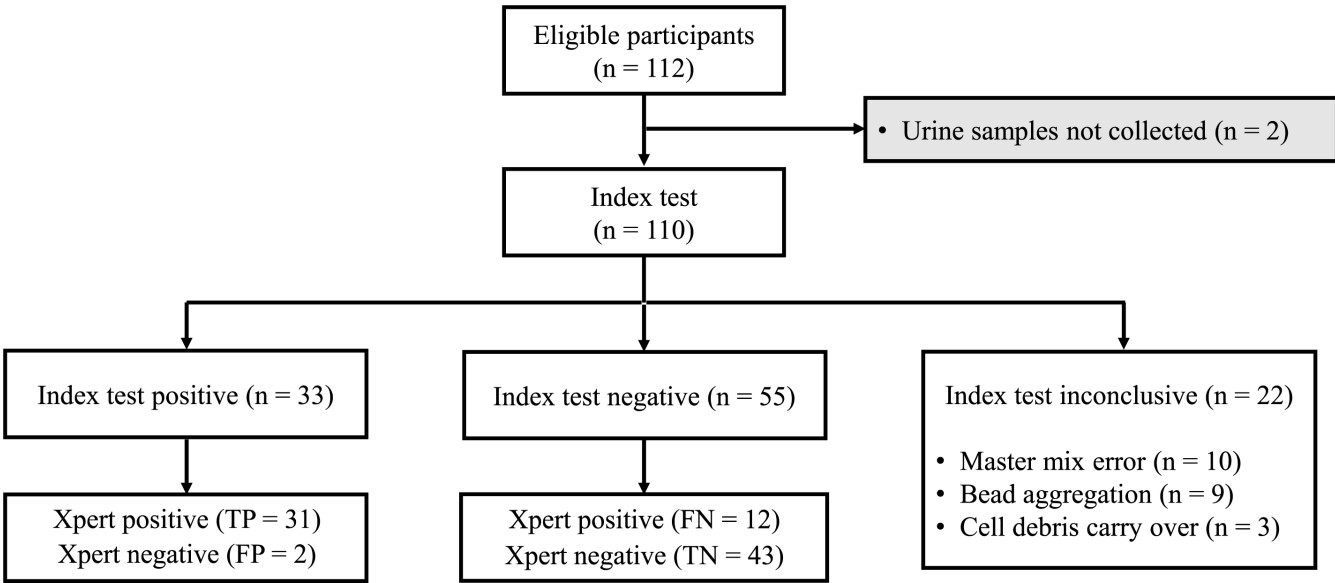

**FIG 2** STARD diagram reporting flow of participants through the study. One hundred twelve participants were recruited, and cryopreserved urine specimens were blindly subjected to the index test (positivity threshold at 4.2 copies of target DNA per 10 mL of urine) and valid assessments stratified by the reference test results (sputum Xpert MTB/RIF assay). TP, true positives; FP, false positives; TN, true negatives; FN, false negatives.

72.1% (CI 0.59–0.86) at 95.5% specificity (CI 0.90–1.02) with a positive predictive value of 93.9% and negative predictive value of 78.2% (Fig. 3B).

## Concordance of reference and index test among HHCs

We then stratified the diagnostic accuracy analysis of the index test considering the presence of TB symptoms in combination with the reference standard test results. The

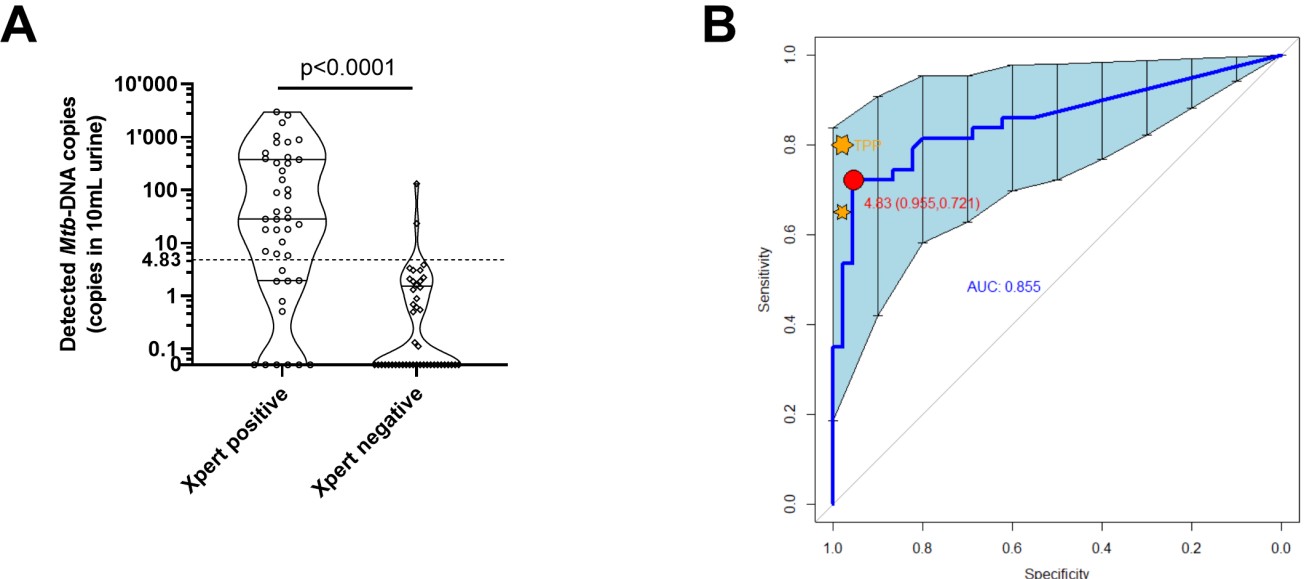

**FIG 3** Index test results' agreement with reference standard. (A) Violin plot with overlaying individual values of *Mtb*-IS6110 DNA copy numbers retrieved from urine specimens of sputum Xpert positive and negative study participants (Mann-Whitney two-tailed test). The dashed horizontal line depicts the optimal cut-off value (Youden's index) determined from the ROC analysis presented in (B) receiving operating characteristic curve with 95% CI. The red dot depicts the optimal sensitivity and specificity of the diagnostic tool based on the Youden's index. The orange stars indicate the optimal and minimal targets for the latest target product profile definition for a non-sputum TB diagnostic test (17).

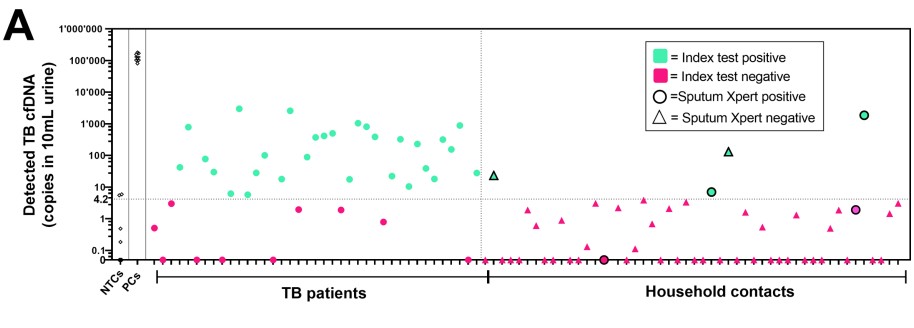

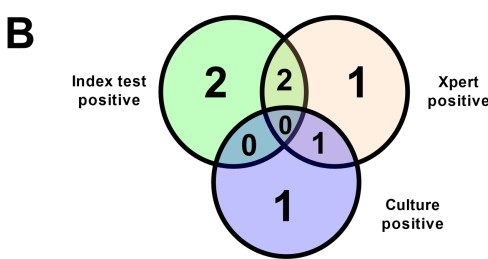

**FIG 4** Index test results stratified by disease status and reference standards. (A) Copy numbers of *Mtb*-IS6110 DNA per 10 mL of urine specimen from sputum Gene Xpert MTB/RIF (Xpert) positive or negative participants. Positive controls (PCs, $10^5$ copies of synthetic DNA target) and negative template controls (NTCs). Sputum GeneXpert MTB/RIF-negatives are depicted as triangles or positives as circles. Index test positives are colored in green and negatives in pink. (B) Venn diagram showing intersections and exclusivities of positive test results among HHCs for the index test, Xpert test, and culture test (clinical characteristics of respective individuals presented in Table 2).

stratification analysis is presented in Fig. 4A, including NTCs and reference positive controls cumulated across seven individual rounds of index test batches. The clinical characteristics of the respective household contacts are listed in Table 2. Four household contacts generated a positive result following sputum Xpert MTB/RIF testing, of which two were concomitantly identified as TB cases by the index test and one by culture. One TB instance was identified solely by culture. The index test identified two additional individuals that were not detected by the sputum Xpert MTB/RIF test nor by culture. The consistency and individuality of the three tests are summarized in Fig. 4B. A follow-up investigation revealed that the two asymptomatic HHCs that displayed a positive index test in the absence of molecular or culture confirmation later developed active TB that was diagnosed at National Tuberculosis and Leprosy Program laboratory by sputum Xpert MTB/RIF testing and received treatment (Table 2). Eventually, all "false positive" index tests turned out to be true asymptomatic TB infection cases that later required TB treatment.

**TABLE 2** HHCs with discrepant index test, GeneXpert MTB/RIF (Xpert) test, or culture test (LJ) results: clinical characteristics at enrolment and follow-up visits

|  | | TB test | | | | |
|  | Urine | Sputum | | | | |
| HHCs | Index | Xpert | LJ | Sex | Symptom | Follow-up, treatment, and outcome |
|---|---|---|---|---|---|---|
| I | + | − | − | M | None | Completed TB treatment in August 2022 |
| II | + | − | − | M | None | Completed TB treatment in September 2022 |
| III | + | + | − | M | None | Completed TB treatment in September 2022 |
| IV | + | + | − | M | None | Lost on follow-up |
| V | − | + | − | M | Cough | Completed TB treatment in September 2022 |
| VI | − | + | + | F | None | Completed TB treatment in November 2022, currently complaining of general joint pain |
| VII | − | − | + | F | None | Initiated TB treatment in February 2023 |

## DISCUSSION

In a case-control study design, we reported here that a magnetic bead-capture assay based on urine specimens could discriminate active TB from sputum Xpert MTB/RIF negative household contacts with an overall sensitivity of 72.1% and specificity of 95.5%. In addition, we showed that all false-positive index tests had identified subclinical TB cases. The likelihood that individuals being asymptomatically infected with *Mtb* may be contributing to the transmission of TB has been neglected in the past but is now increasingly gaining attention (5, 6). Available TB detection methods are commonly used on symptomatic individuals and optimized for the detection of active pulmonary forms of the disease. New TB diagnostic tools that are independent of sputum collection and suitable for community-wide screening are urgently needed and should include the detection of subclinical forms of TB.

Most diagnostic methods that are already approved or in development are based on clinical specimens, for which the collection can be invasive and/or technically challenging, e.g., sputum, blood, nasal swabs, or breath condensates. Collecting such specimens from asymptomatic individuals appears both inconvenient due to the need to visit a healthcare facility and/or impossible due to the inability to produce sputum. Although blood previously displayed high diagnostic accuracy in our study setting (20, 21), we anticipated that fear and inconvenience of giving blood are likely restraining healthy individuals from participating in a diagnostic study (37). In contrast, urine constitutes a particularly convenient non-invasive sample alternative that can be collected without the need for specialized equipment or trained personnel (32). Currently endorsed TB diagnostic tools based on urine only detect the presence of LAM, a glycolipid shed from the mycobacterial cell envelope. Overall, LAM-based lateral-flow assays displayed limited sensitivity, likely due to the dilution effect of the large volume inherent to urine specimens (38). In that context, Oreskovic et al. developed a molecular hybridization assay to concentrate *Mtb*-DNA targets from urine providing good accuracy and offering a promising alternative to current TB diagnostics (32).

With a sensitivity of 72.1% and a specificity of 95.6%, the implementation of Oreskovic's hybridization assay closely meets the TPP defined by the WHO set at 65% and 98%, respectively (17). Our preliminary assessment confirmed that even in the presence of DNA preservatives, urine specimens should be processed as soon as possible or otherwise stored at −80°C to slow down DNA degradation occurring within minutes at room temperature (Supplemental Material) (39, 40). Therefore, the sensitivity of the index test will likely further improve in a real-time clinical setting where urine specimens will be processed directly to overcome the degradation of the short DNA fragments in urine (41). Furthermore, following the concept of diagnostic "yield," even if the sensitivity is lower than Xpert MTB/RIF on sputum, the yield of the index test could be higher due to better availability of urine specimens in general (42). Moreover, the specificity of a diagnostic test refers to its ability to accurately identify individuals without the specific condition of interest, i.e., the true negatives. False positives are not necessarily artefacts and may originate from an imperfect sensitivity of the gold standard that leads to an underestimation of true positives. In the context of community screening, false positives are particularly interesting as they may reveal cases of asymptomatic TB. The identification of asymptomatic or subclinical TB cases requires cross-sectional surveys or thorough examination of individuals at risk that may include follow-up clinical investigations. In our study, four household contacts displayed positive index test results in the absence of TB symptoms. With only a partial overlap, four household contacts also displayed positive results for the reference standard, among which, one already reported cough at the time of enrolment and another one showed a positive result upon culture testing. Cross-comparison revealed that only two household contacts had positive results for both index and reference tests. Follow-up clinical investigations among these four index-test false-positive participants indicated that three developed active TB subsequently and underwent TB treatment, while the fourth was lost on follow-up yet had displayed a positive sputum Xpert MTB/RIF result. In summary, 100% of the supposedly

"false positive" index tests turned out to be asymptomatic TB infections. We solely performed follow-up investigations of household contacts with a positive index test as a TB episode occurring later among index test negative controls would rather point toward a new infection than progression from a subclinical infection. The discrepancy between Xpert MTB/RIF and urine testing in instances where only the index test delivered positive results may be explained by the fact that during asymptomatic TB, mycobacteria may subsist within interstitial tissue and not yet have reached the airways yet still raising *Mtb* cell-free DNA access to the urine through the lymph/blood/bladder circuit. Taken together, our data show that the urine-based index test identifies household contacts with asymptomatic TB in a complementary manner and at least comparable sensitivity to sputum Xpert MTB/RIF testing.

Our study has two main limitations. First, our sample size was relatively small and as a consequence, our results represent a proof-of-concept that requires further validation across independent study sites. Second, we had to exclude index test results from 22 participants (20%), which reflects the technical complexity of this methodology and that would require further simplification for it to be implemented routinely. After technical errors, bead aggregation was identified as the main reason for the exclusion of assay results, and this may be due to the addition of streptavidin-coated capture beads onto urine samples naturally containing high amounts of biotin. To avoid this, biotin depletion via ultrafiltration or precipitation may be tested prior adding the streptavidin-coated beads. Alternatively, a different coupling method could be used to generate the capture beads. Moreover, TB caused by *Mtb* strains harboring no or only a few copies of IS6110 may lead to false negative results. Further analysis including genome sequencing of the infecting strains could estimate the impact of IS6110 genome copy numbers on the index test accuracy across different study settings. Along this line, Oreskovic et al. reported a median of 150 *Mtb*-DNA copies per 10 mL urine collected from TB patients in South Africa, while we reported a median of 30 copies in our cohort of Tanzanian TB patients. In the Tanzanian context, lineage 1 strains may harbor lower numbers of IS6110 copies compared to other phylogenetic lineages of *Mtb* (43). Lineage 1 strains are more prevalent in Tanzania (16.5%) than in South Africa (1.75%), and this could have contributed to a generally lower recovery of *Mtb*-DNA targets from the specimen of Tanzanian TB patients (44, 45). In the future, optimization of a multiplexed assay could make use of existing platforms to develop a cartridge-based approach pre-loaded with capture beads prior molecular amplification and detection of several independent *Mtb*-DNA targets to increase sensitivity.

In conclusion, we show that *Mtb*-DNA hybridization assays on urine specimens have the potential to accurately detect TB infections, and we further demonstrate their capacity to detect cases of subclinical TB. With *Mtb*-DNA being found in urine samples from children and individuals with extrapulmonary TB as well as patients co-infected with HIV, the assay could fill the diagnostic gap for these specific patient groups (24, 46, 47). Importantly, urine samples can be pooled easily, rendering such tests particularly suitable for prospective community-wide screening.

## AUTHOR AFFILIATIONS

[1]Swiss Tropical and Public Health Institute, Allschwil, Switzerland

[2]University of Basel, Basel, Switzerland

[3]Ifakara Health Institute, Dar es Salaam, Tanzania

[4]School of Life Sciences, Ecole Polytechnique Federale de Lausanne, Lausanne, Switzerland

[5]Swiss Institute of Bioinformatics, Lausanne, Switzerland

[6]Precision Medicine Unit, Lausanne University Hospital and University of Lausanne, Lausanne, Switzerland

## AUTHOR ORCIDs

Sébastien Gagneux  http://orcid.org/0000-0001-7783-9048
Damien Portevin  http://orcid.org/0000-0003-2949-9557

## FUNDING

| Funder | Grant(s) | Author(s) |
| --- | --- | --- |
| Swiss National Science Foundation (SNSF) | 177163 | Jacques Fellay |
| | | Damien Portevin |
| | | Klaus Reither |
| | | Sébastien Gagneux |
| Swiss National Science Foundation (SNSF) | 197838 | Damien Portevin |

## ADDITIONAL FILES

The following material is available online.

### Supplemental Material

**Supplemental material (Spectrum00426-24-s0001.docx).** Fig. S1.

### Open Peer Review

**PEER REVIEW HISTORY (review-history.pdf).** An accounting of the reviewer comments and feedback.

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
