## [Reviewer comments · Microbiology Spectrum]

Microbiology Spectrum

Diagnostic accuracy of a sequence-specific *Mtb*-DNA hybridization assay in urine: a case-control study including subclinical TB cases

Yves Tschan, Mohamed Sasamalo, Hellen Hiza, Jacques Fellay, Sébastien Gagneux, Klaus Reither, Jerry Hella, and Damien Portevin

Corresponding Author(s): Damien Portevin, Schweizerisches Tropen- und Public Health-Institut

Review Timeline:

Submission Date:	February 15, 2024
Editorial Decision:	March 14, 2024
Revision Received:	March 27, 2024
Editorial Decision:	April 3, 2024
Revision Received:	April 11, 2024
Editorial Decision:	April 15, 2024
Revision Received:	April 16, 2024
Accepted:	April 19, 2024

Editor: Siu-Kei Chow

Reviewer(s): The reviewers have opted to remain anonymous.

Transaction Report:

DOI: <https://doi.org/10.1128/spectrum.00426-24>

Re: Spectrum00426-24 (Diagnostic accuracy of a sequence-specific *Mtb*-DNA hybridization assay in urine: a case-control study including subclinical TB cases)

Dear Dr. Damien Portevin:

Thank you for the privilege of reviewing your work. Below you will find my comments, instructions from the Spectrum editorial office, and the reviewer comments.

Revision Guidelines

Sincerely,
Siu-Kei Chow
Editor
Microbiology Spectrum

Reviewer #1 (Comments for the Author):

Tschan et al explore the use of urine for the diagnosis of TB. This is a well written paper which addresses an important topic in global health. I enjoyed reading it and applaud the researchers on their efforts. My biggest concern is that they reached their Ct value cutoff for positivity (38.91) by analyzing only 7 negative urine samples. This is simply not enough. Based on my experience the use of a Ct value cutoff of 38.91 is exceedingly high and dangerous, ie can easily lead to false positives. Importantly, based on my understanding, the 7 negative urine samples were not analyzed contemporaneously with positive

samples. This is a recipe for disaster since in real life, positive and negative samples will be intermixed, increasing chances of cross-contamination during bead capture/qRTPCT. Of note, other groups have had to use strict contamination-preventing protocols, including the use of plate covers, to specifically prevent this: PMIDs 26701778, 28923054.

I recommend the authors confirm or disprove my hypothesis by doing the following experiment: perform the pipeline on alternating high-positive and negative urine samples (ie Sample 1. Positive, 2. Negative, 3. Positive, 4. Negative, etc) in which the synthetic positives correspond to the highest level of Mtb DNA you've found in your dataset. We used to do this on 48 negatives and 48 positives in 96 well format to obtain the Ct value and EPF cutoffs to assess positivity (usually with two independent operators). It's important the samples are done all at the same time (and positive and negative samples are done right next to each other) and *exactly* in the same way as the usual protocol, no special precautions should be taken. If all goes perfectly you should get a checkerboard readout for qRTPCR results in which it reads: positive, negative, positive, negative, positive etc. Based on my experience, I'm fairly sure that with the 38.91 cut-off it will lead to several false-positives in this assay; but if not at least you'll know for sure and if yes at least you'll have better cutoff values for future experiments.

One other thing, I was thoroughly confused as to when Xpert results pertained to sputum or urine (was Xpert performed on urine?). Please take extra precautions so that this is crystal clear in your manuscript.

What happened to the other HCs -- did any of them who were negative by all tests develop TB? Also while you mention that 4 HCs positive by the index went on to be treated for TB, 3 subjects who went on to develop TB were negative by the index as well. Not sure if this fact is mentioned anywhere in the paper.

I just need to point out that in the introduction the authors make it a point that urine is ideal for patients who are asymptomatic or who are hard to diagnose/who do not produce sputum. Why not test it on children or HIV+ individuals that are specifically mentioned in the paper?

Minor comments

- 1) Explicitly state whether the method can distinguish between Mtb versus NTM, and what the capture probes target
- 2) Was LAM done on the urine or IGRA done on HC? (A shame if it wasn't)
- 3) No mention is made of LTBI, is this not pertinent (ie couldn't some patients form LTBI and this may explain why they were positive yet asymptomatic)?
- 4) Note, IS6110 is absent from some Beijing strains so the test would be less effective in regions where this strain is common (PMID: 22808061)
- 5) I'm curious, when urine was collected and specific fraction of urine was used for this study (first urine in the morning? and first fraction or mid-stream?) and why limit it to 10ml for the bead capture method from urine? Patients typically produce much more.
- 6) I'd like to know more about the 'synthetic positive control (SPC) template' and how contamination from SPC was prevented
- 7) Please describe precisely what criteria (besides Ct value of 38.91) was used to assess positivity (ie EPF etc).
- 8) Was there any association with Xpert Mtb level vs detection by the index method, or for that matter for any other patient characteristics? It is curious that the index method did not detect the 2 subjects who were culture positive in Fig 4b.
- 9) Almost nothing is ever processed directly in a clinical lab so not realistic: "Therefore, the sensitivity of the index test will likely further improve in a real-time clinical setting where urine specimen will be processed directly to overcome the degradation of the short DNA fragments in urine"
- 10) Please add as supplemental data: "Our preliminary assessment confirmed that even in the presence of DNA preservatives, urine specimen should be processed as soon as possible or otherwise stored at -80{degree sign}C to slow down DNA degradation occurring within minutes at room temperature (data not shown)"

I also just refer the authors to a paper (PMID 30721243) that examines various preservation methods for cell-free DNA in urine and takes another approach to enrich such nucleic acids from large-volume urine samples. Of note, the referenced paper mentions that adding BSA directly to urine helped improve assay performance, though I wonder whether incubating your beads with BSA prior to addition to urine may help with aggregation problems (a real weakness of the current method, ie ~20% failure rate). I recommend the authors do some pilot experiments in this realm, ie can they reduce aggregation failures and expand stability so that the urine does not need to go directly into a -80C freezer, which is basically non-existent in the low-resource areas where TB is most common.

Reviewer #2 (Comments for the Author):

The authors describe a molecular assay to detect *M. tuberculosis* DNA in urine. The intended use for this would be surveillance, especially in exposure investigations of household contacts. The authors thoroughly discuss the limitations and acknowledge that the complexity of the test method would not be easily deployed in the clinical lab space, though this is a proof of concept paper.

- Under the index test adapted in the material and methods, does cooled transport mean refrigerated or frozen?
- Recognizing that this is a proof of principle study, would the authors please comment in the discussion regarding treatment status of the patients, HOCs? It is possible that the Xpert negatives in Figure 4a are due to treatment. The Xpert test is not recommended for use after treatment initiation or for monitoring treatment effectiveness.
- Ensure that bacterial names are italicized throughout- bottom of the second discussion page *Mtb*

Reviewer #1 (Comments for the Author):

Tschan et al explore the use of urine for the diagnosis of TB. This is a well written paper which addresses an important topic in global health. I enjoyed reading it and applaud the researchers on their efforts.

My biggest concern is that they reached their Ct value cutoff for positivity (38.91) by analyzing only 7 negative urine samples. This is simply not enough. Based on my experience the use of a Ct value cutoff of 38.91 is exceedingly high and dangerous, ie can easily lead to false positives. Importantly, based on my understanding, the 7 negative urine samples were not analyzed contemporaneously with positive samples. This is a recipe for disaster since in real life, positive and negative samples will be intermixed, increasing chances of cross-contamination during bead capture/qRTPCT. Of note, other groups have had to use strict contamination-preventing protocols, including the use of plate covers, to specifically prevent this: PMIDs 26701778, 28923054. I recommend the authors confirm or disprove my hypothesis by doing the following experiment: perform the pipeline on alternating high-positive and negative urine samples (ie Sample 1. Positive, 2. Negative, 3. Positive, 4. Negative, etc) in which the synthetic positives correspond to the highest level of Mtb DNA you've found in your dataset. We used to do this on 48 negatives and 48 positives in 96 well format to obtain the Ct value and EPF cutoffs to assess positivity (usually with two independent operators). It's important the samples are done all at the same time (and positive and negative samples are done right next to each other) and *exactly* in the same way as the usual protocol, no special precautions should be taken. If all goes perfectly you should get a checkerboard readout for qRTPCR results in which it reads: positive, negative, positive, negative, positive etc. Based on my experience, I'm fairly sure that with the 38.91 cut-off it will lead to several false-positives in this assay; but if not at least you'll know for sure and if yes at least you'll have better cutoff values for future experiments.

Answer: We sincerely apologize for the misunderstanding. The depicted cut-off in Figure 1b does not constitute the threshold that was applied for accuracy analysis of the index test. Ultimately the index test threshold was issued from the Receiver Operating Curve analysis. Importantly and as highlighted by the reviewer, we did include negative template controls within each run of patient sample processing. This important point has now been further highlighted in the result section of the revised version of the report. (Line 146-148)

One other thing, I was thoroughly confused as to when Xpert results pertained to sputum or urine (was Xpert performed on urine?). Please take extra precautions so that this is crystal clear in your manuscript.

Answer: We stated in the methods section that "Xpert MTB/RIF on sputum specimen was used as a reference standard " and also in the STARD diagram that the reference test was a "sputum Xpert MTB/RIF assay". Nevertheless, we appreciate that referring to this test throughout the manuscript without specifying the sample origin may be confusing and we would like to thank the reviewer for pointing this out. We have now further highlighted that Xpert testing was systematically performed on sputum in several missing instances (In the abstract as well as line 71, 171, 175 and, 180)

What happened to the other HCs -- did any of them who were negative by all tests develop TB?

Answer: We solely performed follow-up investigations of index test positive TB controls as a TB episode occurring later among index test negative controls would rather point towards a new infection than progression from a subclinical infection. This point is now discussed on lines 230-232.

Also while you mention that 4 HCs positive by the index went on to be treated for TB, 3 subjects who went on to develop TB were negative by the index as well. Not sure if this fact is mentioned anywhere in the paper.

Answer: Thank you again for pointing this out as we had left an error in the original version of the manuscript. The results section has been corrected to read like: "Four (and not three) study participants generated a positive result following sputum Xpert MTB/RIF test result, of which two were also identified as TB cases by the index test and one by culture."

I just need to point out that in the introduction the authors make it a point that urine is ideal for patients who are asymptomatic or who are hard to diagnose/who do not produce sputum. Why not test it on children or HIV+ individuals that are specifically mentioned in the paper?

Answer: We intend to report here a proof-of-concept" study demonstrating that the index test is particularly suitable and relevant for subclinical infection for which sputum collection is inherently challenging. Nevertheless, further studies applied to childhood TB and PLWH are warranted as stated in the conclusion of the manuscript.

Minor comments

1) Explicitly state whether the method can distinguish between Mtb versus NTM, and what the capture probes target

Answer: We stated on line 50 that the assay is "*Mtb*-specific". We have now specified the targeted sequence between brackets.

2) Was LAM done on the urine or IGRA done on HC? (A shame if it wasn't)

Answer: Sorry, we did not perform head-to-head comparison of all other relevant diagnostic platforms. This clinical study specifically aims to report the diagnostic accuracy of the specified index test.

3) No mention is made of LTBI, is this not pertinent (ie couldn't some patients form LTBI and this may explain why they were positive yet asymptomatic)?

Answer: LTBI is indeed no longer a relevant concept. Infection shall now be stratified as symptomatic or not and infectious or not. [https://doi.org/10.1016/S2213-2600\(24\)00028-6](https://doi.org/10.1016/S2213-2600(24)00028-6)

4) Note, IS6110 is absent from some Beijing strains so the test would be less effective in regions where this strain is common (PMID: 22808061)

Answer: This limitation is also true for Lineage 1 strains and discussed on lines 251-253.

5) I'm curious, when urine was collected and specific fraction of urine was used for this study (first urine in the morning? and first fraction or mid-stream?) and why limit it to 10ml for the bead capture method from urine? Patients typically produce much more.

Answer: The urine was collected during the day at the clinic within a collection pot and 10ml transferred in a tube pre-filled with an adequate volume of EDTA + Tris-HCL for DNA preservation.

6) I'd like to know more about the 'synthetic positive control (SPC) template' and how contamination from SPC was prevented.

Answer: Spiking of SPC in controlled urine sample was done in the microbiology lab while patients' samples were thawing. Then all urine samples were centrifugated and processed in parallel for transfer of supernatants into fresh tubes and addition of beads.

7) Please describe precisely what criteria (besides Ct value of 38.91) was used to assess positivity (ie EPF etc).

Answer: The index test threshold was issued from the Receiver Operating Curve analysis

8) Was there any association with Xpert Mtb level vs detection by the index method, or for that matter for any other patient characteristics? It is curious that the index method did not detect the 2 subjects who were culture positive in Fig 4b.

Answer: The sputum Xpert test was performed at the National Tuberculosis and Leprosy program laboratory and then patient offered to enroll the study. By default, we did not have access to the initial Xpert test results.

9) Almost nothing is ever processed directly in a clinical lab so not realistic: "Therefore, the sensitivity of the index test will likely further improve in a real-time clinical setting where urine specimen will be processed directly to overcome the degradation of the short DNA fragments in urine"

Answer: In this proof-of-concept study, urine specimen were cryopreserved up to several weeks before cryo-shipment overseas for extraction and analysis. Even though, the addition of buffer and EDTA greatly slowed DNA degradation it cannot completely prevent it (see supplementary data presented in minor point 10). As such, short-term processing would undoubtedly increase the assay sensitivity.

10) Please add as supplemental data: "Our preliminary assessment confirmed that even in the presence of DNA preservatives, urine specimen should be processed as soon as possible or otherwise stored at -80{degree sign}C to slow down DNA degradation occurring within minutes at room temperature (data not shown)"

Answer: The respective data are now presented as supplementary material.

I also just refer the authors to a paper (PMID 30721243) that examines various preservation methods for cell-free DNA in urine and takes another approach to enrich such nucleic acids from large-volume urine samples. Of note, the referenced paper mentions that adding BSA directly to urine helped improve assay performance, though I wonder whether incubating your beads with BSA prior to addition to urine may help with aggregation problems (a real weakness of the current method, ie ~20% failure rate). I recommend the authors do some pilot experiments in this realm, ie can they reduce aggregation failures and expand stability so that the urine does not need to go directly into a -80C freezer, which is basically non-existent in the low-resource areas where TB is most common.

Answer: Thank you we will certainly consider and test this recommendation in future studies.

Reviewer #2 (Comments for the Author):

The authors describe a molecular assay to detect *M. tuberculosis* DNA in urine. The intended use for this would be surveillance, especially in exposure investigations of household contacts. The authors thoroughly discuss the limitations and acknowledge that the complexity of the test method would not be easily deployed in the clinical lab space, though this is a proof of concept paper.

- Under the index test adapted in the material and methods, does cooled transport mean refrigerated or frozen?

Answer: We meant 4°C, this point is now clarified in the methods section of the revised manuscript.

-Recognizing that this is a proof of principle study, would the authors please comment in the discussion regarding treatment status of the patients, HOCs? It is possible that the Xpert negatives in Figure 4a are due to treatment. The Xpert test is not recommended for use after treatment initiation or for monitoring treatment effectiveness.

Answer: Thank you for pointing this important missing information. Patients were enrolled at the time of diagnosis and specimen collected before initiation of treatment. This point is further clarified in the methods section of the revised manuscript.

-Ensure that bacterial names are italicized throughout- bottom of the second discussion page *Mtb*

Answer: Only instances referring to the proprietary Xpert MTB/RIF branding were kept as such and all other instances are italicized.

Re: Spectrum00426-24R1 (Diagnostic accuracy of a sequence-specific *Mtb*-DNA hybridization assay in urine: a case-control study including subclinical TB cases)

Dear Dr. Damien Portevin:

Thank you for the privilege of reviewing your work. Below you will find the instructions from the Spectrum editorial office, and additional reviewer comments.

An independent third review was performed. Please address the comments/concerns adequately.

Revision Guidelines

Sincerely,
Siu-Kei Chow
Editor
Microbiology Spectrum

Reviewer #3 (Comments for the Author):

The authors evaluated the performance of a hybridization assay to diagnose tuberculosis using urine specimens.

Major comments

1. What was the rationale for rerunning sputum samples after the urine hybridization test? Why not using the initial sputum Xpert

test results? Did HCs receive any sputum Xpert tests before the urine test?

2. The math between figure 2 and the text was not quite right. Lines 158-159 stated "... 39 symptomatic Xpert... and 49 HCs..." Lines 155-156 stated "14 Xpert positive ... and 8 HC samples" were excluded. (I) Doing the math reversely, 39 (included) + 14 (excluded) = 53. However, line 154 stated "... included 52 samples". Does that mean the discrepancy of 1 patient was because his/her urine was not collected? If so, that means in the HC group there was another patient (n=1) whose urine was not collected. (II) 59 (total HC) - 8 (excluded) = 51, which is in contrast to line 159 "...49 HCs". Also, even excluding the patient without urine collection, the result is 50, but not 49. Please review the data in the text and figure 2, and clarify the math/number of subjects.

3. The presentation of figure 2 was unclear. Was the urine test positive, Xpert negative (n=2) from HCs? How many of the urine test negative, Xpert positive (n=12) belonged to original Xpert positive?

4. I strongly recommend figure 2 be revised into a different workflow

Eligible (n=112) index test (n=110) initial Xpert positive (n = 52) VS HC (n = 58)

initial Xpert positive (n = 52) excluded (n = 14), include reasons if needed VS urine positive (n=?) VS negative (n = 52 - 14 - ?)

HC (n=58) excluded (n = 8), include reasons if needed VS HC urine positive (n=??) VS HC negative (n = 58 - 8 - ??)

HC urine positive (after sputum testing) Sputum Xpert positive VS negative

HC urine negative (after sputum testing) Sputum Xpert positive VS negative

5. Figure 3A showed 2 samples with Xpert negative/urine positive (2 triangles under HCs in figure 4A). Were the Xpert tests repeated? How would you explain the discrepancy? Please elaborate in the manuscript.

6. Please include a new 2x2 table to illustrate the performance (true pos, false pos, true neg, false neg) to help readers better understand the calculation of sensitivity and specificity. If the 2 samples mentioned above in #5 could be rerun on Xpert, an adjusted table could be included as well.

7. Lines 172-174 was long and confusing. Were "Four study participants" from HCs? Please break down the sentence to help readers understand better. "...generated a positive result... following Xpert test result". What test result after Xpert result? The sequence and the type of test were not clear.

8. The whole manuscript was focusing on Xpert vs urine test. The inclusion of culture (table 1) has further complicated the study, let alone the ambiguity of figure 2 (flowchart). Table 2 showed HC VII presented with results of urine negative, Xpert negative, culture positive, indicating that culture results in this study could be different compared to the 2 main molecular methods. In fact, table 1 showed n = 23 positive for MTBC, whereas 53 were positive for MTB Xpert. Given the fact that culture is used as the gold standard, what was the cause of this discrepancy? This discrepancy as well as the overall culture results in all patients need to be elaborated.

Minor comments

1. Abstract and throughout the manuscript: the term "symptomatic sputum Xpert MTB/RIF positive TB patients" should be revised. It is grammatically wrong.

2. Abstract: "discriminate active TB from Xpert MTB/RIF negative household contacts..." was unclear. Please rephrase.

3. Please remove "proof-of-concept" in the manuscript. It does not add value and is distracting.

4. Lines 69-70: prior to the initiation

5. Lines 100-101: The sentence "specimens were incubated at 120C for 15 minutes" was unclear. The denature step setup does not usually go over 100C as the boiling point. Please revise.

6. Please standardize the units in the manuscript. For examples, min vs minute

7. What is "NTLP" laboratory (line179)? Please clarify in the text.

8. Consider moving part of lines 226-234 to the result section to help explain the discrepancies and follow-up.

9. Footnote in table 1: spell out Rif. MTBC* and NTM were incorrectly formatted.

Reviewer #3 (Comments for the Author):

Major comments

1. What was the rationale for rerunning sputum samples after the urine hybridization test? Why not using the initial sputum Xpert test results? Did HCs receive any sputum Xpert tests before the urine test?

Answer: We did not "rerun" sputum samples after the urine hybridization test. As described in the Material and Methods and Reference standard section, both patients and household contacts underwent Xpert testing at the time of enrolment.

2. The math between figure 2 and the text was not quite right. Lines 158-159 stated "... 39 symptomatic Xpert... and 49 HCs..." Lines 155-156 stated "14 Xpert positive ... and 8 HC samples" were excluded. (I) Doing the math reversely, 39 (included) + 14 (excluded) = 53 . However, line 154 stated "... included 52 samples". Does that mean the discrepancy of 1 patient was because his/her urine was not collected? If so, that means in the HC group there was another patient (n=1) whose urine was not collected. (II) 59 (total HC) - 8 (excluded) = 51, which is in contrast to line 159 "...49 HCs". Also, even excluding the patient without urine collection, the result is 50, but not 49. Please review the data in the text and figure 2, and clarify the math/number of subjects.

Answer: Thank you for pointing out this error, Lines 155-156 should read "13 Xpert positive ... and 9 HC samples". The text has been corrected accordingly in the revised version of the manuscript.

3. The presentation of figure 2 was unclear. Was the urine test positive, Xpert negative (n=2) from HCs? How many of the urine test negative, Xpert positive (n=12) belonged to original Xpert positive?

Answer: The presentation of figure 2 strictly follows STARD guidelines to reflect how the index test performed when compared to the reference standard. However, figure 4A is aimed at answering this specific question and show the two tests performed across the two arms (patients versus contacts). The answer to your question is represented by the lower left quadrant of figure 4A so n=10. We are now referring to Figure 4A line 158 of the revised manuscript to clarify this further.

4. I strongly recommend figure 2 be revised into a different workflow

Eligible (n=112) \diamond index test (n=110) \diamond initial Xpert positive (n = 52) VS HC (n = 58)

initial Xpert positive (n = 52) \diamond excluded (n = 14), include reasons if needed VS urine positive (n=?) VS negative (n = 52 - 14 - ?)

HC (n=58) \diamond excluded (n = 8), include reasons if needed VS HC urine positive (n=??) VS HC negative (n = 58 - 8 - ??)

HC urine positive (after sputum testing) \diamond Sputum Xpert positive VS negative

HC urine negative (after sputum testing) \diamond Sputum Xpert positive VS negative

Answer: Our study diagram is strictly adhering to the latest standard (STARD 2015 guidelines) for reporting diagnostic accuracy studies, see following link to prototype diagram.

(<https://www.equator-network.org/wp-content/uploads/2015/03/STARD-2015-flow-diagram.pdf>).

We have relabeled the respective title legend to emphasize this important aspect.

5. Figure 3A showed 2 samples with Xpert negative/urine positive (2 triangles under HCs in figure

4A). Were the Xpert tests repeated? How would you explain the discrepancy? Please elaborate in the manuscript.

Answer: These two samples are also highlighted in Figure 4B Venn diagram (Upper left green section). Xpert was not repeated yet the respective individuals subjected to follow-up investigations detailed in Table 2 (Individuals I and II). The discrepancy between Xpert MTB/RIF and urine testing in instances where only the index test delivered positive results may be explained by the fact that during asymptomatic TB, mycobacteria may subsist within interstitial tissue and not yet have reached the airways still raising *Mtb* cell free DNA access to the urine through the lymph/blood/bladder circuit. This point has been elaborated on lines 235-239 of the revised manuscript.

6. Please include a new 2x2 table to illustrate the performance (true pos, false pos, true neg, false neg) to help readers better understand the calculation of sensitivity and specificity. If the 2 samples mentioned above in #5 could be rerun on Xpert, an adjusted table could be included as well.

Answer: The requested 2*2 table is actually embedded within the two lower left quadrants of the STARD diagram presented in Figure 2. To help the readers, we have now specified (TP, FP, TN & FN) in the revised version of the figure. We did not store sputum samples and cannot rerun Xpert test retrospectively nevertheless our follow-up investigations confirmed TB diagnosis for the 2 false positive instances. We emphasized strongly that point in the abstract stating that: "The detection of *Mtb*-specific DNA in urine suggested four asymptomatic TB infection cases that were confirmed in all instances either by concomitant Xpert MTB/RIF sputum testing or by follow-up investigation raising the specificity of the index test to 100%."

7. Lines 172-174 was long and confusing. Were "Four study participants" from HCs? Please break down the sentence to help readers understand better. "...generated a positive result... following Xpert test result". What test result after Xpert result? The sequence and the type of test were not clear.

Answer: We apologize for leaving this error behind. The sentence has been corrected as follow: "Four household contacts generated a positive result following sputum Xpert MTB/RIF testing, of which two were concomitantly identified as TB cases by the index test and one by culture."

8. The whole manuscript was focusing on Xpert vs urine test. The inclusion of culture (table 1) has further complicated the study, let alone the ambiguity of figure 2 (flowchart). Table 2 showed HC VII presented with results of urine negative, Xpert negative, culture positive, indicating that culture results in this study could be different compared to the 2 main molecular methods. In fact, table 1 showed n = 23 positive for MTBC, whereas 53 were positive for MTB Xpert. Given the fact that culture is used as the gold standard, what was the cause of this discrepancy? This discrepancy as well as the overall culture results in all patients need to be elaborated.

Answer: Our laboratory routinely perform culture testing to isolate *Mtb* strains for epidemiological studies. We reported culture results here for the sake of transparency. Culture does not constitute our gold standard, our Tanzanian patients did not receive treatment based on culture results. Our patients were diagnosed and treated based on Xpert testing and enrolled in our study accordingly and as reported in the Methods section.

Minor comments

1. Abstract and throughout the manuscript: the term "symptomatic sputum Xpert MTB/RIF positive TB patients" should be revised. It is grammatically wrong.

Answer: Agreed and rephrased as "TB patients"

2. Abstract: "discriminate active TB from Xpert MTB/RIF negative household contacts..." was unclear. Please rephrase.

Answer: Agreed and rephrased as "discriminate active TB from healthy household contacts"

3. Please remove "proof-of-concept" in the manuscript. It does not add value and is distracting.

Answer: Agreed and removed

4. Lines 69-70: prior to the initiation

Answer: Thank you, corrected.

5. Lines 100-101: The sentence "specimens were incubated at 120C for 15 minutes" was unclear. The denature step setup does not usually go over 100C as the boiling point. Please revise.

Answer: This is not a mistake. The denaturation step is performed in a dry bath so this the necessary time and temperature so that the urine specimen reach a temperature above 90°C. We have added this detail.

6. Please standardize the units in the manuscript. For examples, min vs minute

Answer: Thank you, corrected.

7. What is "NTLP" laboratory (line179)? Please clarify in the text.

Answer: Thank you, clarified as "National Tuberculosis and Leprosy Program".

8. Consider moving part of lines 226-234 to the result section to help explain the discrepancies and follow-up.

Answer: Thank you we now emphasized at the end of the results section that: "Eventually, all "false positive" index tests turned out to be true asymptomatic TB infections cases that later required TB treatment."

9. Footnote in table 1: spell out Rif. MTBC* and NTM were incorrectly formatted

Answer: Thank you, corrected.

Re: Spectrum00426-24R2 (Diagnostic accuracy of a sequence-specific *Mtb*-DNA hybridization assay in urine: a case-control study including subclinical TB cases)

Dear Dr. Damien Portevin:

Thank you for the privilege of reviewing your work. Below you will find my comments, instructions from the Spectrum editorial office, and the reviewer comments.

Multiple revisions indicated in the response letter were missing from the main manuscript. Please make sure the manuscript is revised, proofread, and uploaded appropriately before resubmission.

Revision Guidelines

Sincerely,
Siu-Kei Chow
Editor
Microbiology Spectrum

Reviewer #3 (Comments for the Author):

Major comments

1. What was the rationale for rerunning sputum samples after the urine hybridization test? Why not using the initial sputum Xpert test results? Did HCs receive any sputum Xpert tests before the urine test?

Answer: We did not "rerun" sputum samples after the urine hybridization test. As described in the Material and Methods and Reference standard section, both patients and household contacts underwent Xpert testing at the time of enrolment.

2. The math between figure 2 and the text was not quite right. Lines 158-159 stated "... 39 symptomatic Xpert... and 49 HCs..." Lines 155-156 stated "14 Xpert positive ... and 8 HC samples" were excluded. (I) Doing the math reversely, 39 (included) + 14 (excluded) = 53 . However, line 154 stated "... included 52 samples". Does that mean the discrepancy of 1 patient was because his/her urine was not collected? If so, that means in the HC group there was another patient (n=1) whose urine was not collected. (II) 59 (total HC) - 8 (excluded) = 51, which is in contrast to line 159 "...49 HCs". Also, even excluding the patient without urine collection, the result is 50, but not 49. Please review the data in the text and figure 2, and clarify the math/number of subjects.

Answer: Thank you for pointing out this error, Lines 153-154 should read "13 Xpert positive ... and 9 HC samples". The text has been corrected accordingly in the revised version of the manuscript.

3. The presentation of figure 2 was unclear. Was the urine test positive, Xpert negative (n=2) from HCs? How many of the urine test negative, Xpert positive (n=12) belonged to original Xpert positive?

Answer: The presentation of figure 2 strictly follows STARD guidelines to reflect how the index test performed when compared to the reference standard. However, figure 4A is aimed at answering this specific question and show the two tests performed across the two arms (patients versus contacts). The answer to your question is represented by the lower left quadrant of figure 4A so n=10. We are now referring to Figure 4A line 157 of the revised manuscript to clarify this further.

4. I strongly recommend figure 2 be revised into a different workflow

Eligible (n=112) \diamond index test (n=110) \diamond initial Xpert positive (n = 52) VS HC (n = 58)

initial Xpert positive (n = 52) \diamond excluded (n = 14), include reasons if needed VS urine positive (n=?) VS negative (n = 52 - 14 - ?)

HC (n=58) \diamond excluded (n = 8), include reasons if needed VS HC urine positive (n=??) VS HC negative (n = 58 - 8 - ??)

HC urine positive (after sputum testing) \diamond Sputum Xpert positive VS negative

HC urine negative (after sputum testing) \diamond Sputum Xpert positive VS negative

Answer: Our study diagram is strictly adhering to the latest standard (STARD 2015 guidelines) for reporting diagnostic accuracy studies, see following link to prototype diagram.

<https://www.equator-network.org/wp-content/uploads/2015/03/STARD-2015-flow-diagram.pdf>.

We have relabeled the respective title legend to emphasize this important aspect.

5. Figure 3A showed 2 samples with Xpert negative/urine positive (2 triangles under HCs in figure

4A). Were the Xpert tests repeated? How would you explain the discrepancy? Please elaborate in the manuscript.

Answer: These two samples are also highlighted in Figure 4B Venn diagram (Upper left green section). Xpert was not repeated yet the respective individuals subjected to follow-up investigations detailed in Table 2 (Individuals I and II). The discrepancy between Xpert MTB/RIF and urine testing in instances where only the index test delivered positive results may be explained by the fact that during asymptomatic TB, mycobacteria may subsist within interstitial tissue and not yet have reached the airways still raising *Mtb* cell free DNA access to the urine through the lymph/blood/bladder circuit. This point has been elaborated on lines 235-238 of the revised manuscript.

6. Please include a new 2x2 table to illustrate the performance (true pos, false pos, true neg, false neg) to help readers better understand the calculation of sensitivity and specificity. If the 2 samples mentioned above in #5 could be rerun on Xpert, an adjusted table could be included as well.

Answer: The requested 2*2 table is actually embedded within the two lower left quadrants of the STARD diagram presented in Figure 2. To help the readers, we have now specified (TP, FP, TN & FN) in the revised version of the figure and legend. We did not store sputum samples and cannot rerun Xpert test retrospectively nevertheless our follow-up investigations confirmed TB diagnosis for the 2 false positive instances. We emphasized strongly that point in the abstract stating that: "The detection of *Mtb*-specific DNA in urine suggested four asymptomatic TB infection cases that were confirmed in all instances either by concomitant Xpert MTB/RIF sputum testing or by follow-up investigation raising the specificity of the index test to 100%."

7. Lines 172-174 was long and confusing. Were "Four study participants" from HCs? Please break down the sentence to help readers understand better. "...generated a positive result... following Xpert test result". What test result after Xpert result? The sequence and the type of test were not clear.

Answer: We would like to apologize for leaving this typographic error behind. The sentence has been corrected as follow: "Four household contacts generated a positive result following sputum Xpert MTB/RIF testing, of which two were concomitantly identified as TB cases by the index test and one by culture."

8. The whole manuscript was focusing on Xpert vs urine test. The inclusion of culture (table 1) has further complicated the study, let alone the ambiguity of figure 2 (flowchart). Table 2 showed HC VII presented with results of urine negative, Xpert negative, culture positive, indicating that culture results in this study could be different compared to the 2 main molecular methods. In fact, table 1 showed n = 23 positive for MTBC, whereas 53 were positive for MTB Xpert. Given the fact that culture is used as the gold standard, what was the cause of this discrepancy? This discrepancy as well as the overall culture results in all patients need to be elaborated.

Answer: Our laboratory routinely perform culture testing to isolate *Mtb* strains for epidemiological studies. We reported culture results here for the sake of transparency. Culture does not constitute our gold standard, our Tanzanian patients did not receive treatment based on culture results. Our patients were diagnosed and treated based on Xpert testing and enrolled in our study accordingly and as reported in the Methods section.

Minor comments

1. Abstract and throughout the manuscript: the term "symptomatic sputum Xpert MTB/RIF positive TB patients" should be revised. It is grammatically wrong.

Answer: Agreed and rephrased as "TB patients"

2. Abstract: "discriminate active TB from Xpert MTB/RIF negative household contacts..." was unclear. Please rephrase.

Answer: Agreed and rephrased as "discriminate active TB from healthy household contacts"

3. Please remove "proof-of-concept" in the manuscript. It does not add value and is distracting.

Answer: Agreed and removed

4. Lines 69-70: prior to the initiation

Answer: Thank you, corrected.

5. Lines 100-101: The sentence "specimens were incubated at 120C for 15 minutes" was unclear. The denature step setup does not usually go over 100C as the boiling point. Please revise.

Answer: This is not a mistake. The denaturation step is performed in a dry bath so this the necessary time and temperature so that the urine specimen reach a temperature above 90°C. We have added this detail.

6. Please standardize the units in the manuscript. For examples, min vs minute

Answer: Thank you, corrected.

7. What is "NTLP" laboratory (line179)? Please clarify in the text.

Answer: Thank you, clarified as "National Tuberculosis and Leprosy Program".

8. Consider moving part of lines 226-234 to the result section to help explain the discrepancies and follow-up.

Answer: Thank you we now emphasized at the end of the results section that: "Eventually, all "false positive" index tests turned out to be true asymptomatic TB infections cases that later required TB treatment."

9. Footnote in table 1: spell out Rif. MTBC* and NTM were incorrectly formatted

Answer: Thank you, corrected.

Re: Spectrum00426-24R3 (Diagnostic accuracy of a sequence-specific *Mtb*-DNA hybridization assay in urine: a case-control study including subclinical TB cases)

Dear Dr. Damien Portevin:

Your manuscript has been accepted, and I am forwarding it to the ASM production staff for publication. Your paper will first be checked to make sure all elements meet the technical requirements. ASM staff will contact you if anything needs to be revised before copyediting and production can begin. Otherwise, you will be notified when your proofs are ready to be viewed.

Sincerely,
Siu-Kei Chow
Editor
Microbiology Spectrum